# Marine-Fungus-Derived Natural Compound 4-Hydroxyphenylacetic Acid Induces Autophagy to Exert Antithrombotic Effects in Zebrafish

**DOI:** 10.3390/md22040148

**Published:** 2024-03-27

**Authors:** Shaoshuai Xin, Mengqi Zhang, Peihai Li, Lizhen Wang, Xuanming Zhang, Shanshan Zhang, Zhenqiang Mu, Houwen Lin, Xiaobin Li, Kechun Liu

**Affiliations:** 1Biology Institute, Qilu University of Technology (Shandong Academy of Sciences), Jinan 250103, China; qiluxss@163.com (S.X.); liph@sdas.org (P.L.); wlzh1106@126.com (L.W.); zhangmx@sdas.org (X.Z.); zhangss@sdas.org (S.Z.); 2Key Laboratory of Novel Food Resources Processing, Ministry of Agriculture and Rural Affairs, Key Laboratory of Agro-Products Processing Technology of Shandong Province, Institute of Agro-Food Science and Technology, Shandong Academy of Agricultural Sciences, 23788 Gongye North Road, Jinan 250100, China; mengqi139@126.com; 3Chongqing Key Laboratory of High Active Traditional Chinese Medicine Delivery System & Chongqing Engineering Research Center of Pharmaceutical Sciences, Chongqing Medical and Pharmaceutical College, Chongqing 410331, China; 10976@cqmpc.edu.cn; 4Research Center for Marine Drugs, State Key Laboratory of Oncogenes and Related Genes, Department of Pharmacy, School of Medicine, Shanghai Jiao Tong University, Shanghai 200127, China; franklin67@126.com

**Keywords:** marine-derived fungus, 4-hydroxyphenylacetic acid, antithrombotic, autophagy

## Abstract

Marine natural products are important sources of novel drugs. In this study, we isolated 4-hydroxyphenylacetic acid (HPA) from the marine-derived fungus *Emericellopsis maritima* Y39–2. The antithrombotic activity and mechanism of HPA were reported for the first time. Using a zebrafish model, we found that HPA had a strong antithrombotic activity because it can significantly increase cardiac erythrocytes, blood flow velocity, and heart rate, reduce caudal thrombus, and reverse the inflammatory response caused by Arachidonic Acid (AA). Further transcriptome analysis and qRT–PCR validation demonstrated that HPA may regulate autophagy by inhibiting the PI3K/AKT/mTOR signaling pathway to exert antithrombotic effects.

## 1. Introduction

Thrombus can lead to acute myocardial infarction, ischemic heart disease, valvular heart disease, peripheral vascular disease, and other cardiovascular diseases (CVDs), and approximately 20 million people die of cardiovascular disease each year [1,2]. Due to aging, changing lifestyles, and population expansion, the burden of thrombotic disorders on health care systems is increasing [3]. Existing studies suggest that signaling pathways, such as the NF-κB, Toll’s-like receptor, MAPK, mTOR, PI3K, and autophagy pathways, are associated with thrombosis [4,5,6,7]. Acetylsalicylic acid (ASA), warfarin, and heparin are currently the most widely used antithrombotic medications; nonetheless, they have side effects, including bleeding and drug resistance [8,9]. Therefore, there is an urgent need for new antithrombotic drugs that are safer and more reliable than existing drugs.

Natural products are important sources for the discovery of antithrombotic drugs. The oligotrophic environment, low temperature, high salinity, high pressure, and low oxygen content found in the ocean allow organisms to have special metabolic adaptation mechanisms and produce novel structures and a variety of biological activities [10]. Marine fungi are a rich source of natural products [11]. The fungal genus *Emericellopsis* (Hypocreales, Sordariomycetes, Ascomycota) has been found to comprise 27 species, including 23 species and 4 varieties [12]. In recent years, a significant amount of secondary metabolites, such as peptides, terpenes, peptaibols, alkaloids, and flavonoids, have been isolated from *Emericellopsis* sp. [12,13,14]. To date, research on the bioactivities of secondary metabolites from *Emericellopsis* sp. has focused on their anti-inflammatory, antibacterial, and antitumor activities [12,13,15,16,17,18], but few studies have investigated their antithrombotic activity.

The zebrafish is a small vertebrate tropical water fish that has become a widely used model organism for screening natural active compounds [19,20]. The zebrafish model has the characteristics of high throughput and high connotation and has become an important link between cell and mammalian models in drug screening and evaluation systems. The thrombus formation mechanism of zebrafish is similar to that of humans, whose thrombocytes express the platelet glycoproteins GPIIb/IIIa and GPIb and have both endogenous and exogenous clotting pathways [21]. Experiments have shown that the efficacy of some antithrombotic drugs commonly used in the clinic on zebrafish is similar to that in mammals [22], indicating the reliability of screening antithrombotic drugs in zebrafish models. At the protein level, the homology of key parts of zebrafish and humans is nearly 100%, so zebrafish is also very suitable for the study of signaling pathways and the screening of targeted drugs [23]. AA and phenyl hydrazine (PHZ) are commonly used to construct zebrafish thrombus models for screening and mechanistic studies of antithrombotic active ingredients [24,25].

Previously, using a zebrafish model, we examined the antithrombotic activity of extracts from small-scale fermentations of eleven different strains of marine-derived fungus. The results showed that extracts from the marine-derived fungus *Emericellopsis maritima* Y39–2 have antithrombotic activity. In this study, we focused on isolating biologically active natural products from Y39–2, and the metabolite 4-hydroxyphenylacetic acid (HPA) (Figure 1) was separated and shown to have significant antithrombotic activity. Mechanistic investigations revealed that HPA can inhibit the mTOR signaling pathway to activate autophagy, thus playing an antithrombotic role.

HPA is a phenolic compound that was first isolated from the pathogen *Hypochnus sasakii* [26] and subsequently found in the fungi *Oidiodendron* sp. [27] and *Neofusicoccum parvum* [28], the brown alga *Undaria* [29], colonic microorganisms [30], the Chinese herb *Aster tataricus* [31], and human saliva [32]. This compound has a wide range of effects, such as attenuating sepsis-induced acute kidney injury [30], preventing acute acetaminophen-induced liver injury [33], reducing inflammation and edema in lung injury [31], and regulating growth [29,34], anxiolytic [35], antioxidative [36], antimicrobial [37,38], and nematicidal activity [27]. This is the first time that the antithrombotic activity and mechanism of HPA have been explored. The results obtained in this study broaden the resources of potential antithrombotic agents and lay the foundation for developing new antifungal drugs.

## 2. Results

### 2.1. Safe Concentration of HPA for Zebrafish

We treated larvae 72 h post-fertilization (hpf) with different concentrations of HPA for 6 h. The results are shown in Figure 2. At 6 h post-exposure (hpe), no mortality was detected in zebrafish larvae that were exposed to 0 to 657.2 µM HPA. When the HPA treatment concentration was increased from 1314.5 to 3943.5 µM, the mortality rate of zebrafish larvae increased sharply from 6.67% to 100%. The lethality curves were plotted based on the mortality of zebrafish larvae, and the 1% lethal concentration (LC_1_) for HPA was calculated to be 914.6 µM. No malformations were observed in any of the HPA treatment groups. Based on the test results, concentrations of 82.2, 164.3, 328.6, and 657.2 µM were selected for the evaluation of the antithrombotic activity of HPA.

### 2.2. Antithrombotic Activity of HPA

As shown in Figure 3, the area and staining intensity of caudal thrombus in zebrafish in the model group were obviously greater than those in the control group, indicating that AA can significantly induce caudal thrombus formation. The ASA group and all of the HPA treatment groups showed notable reductions in the area and staining intensity of caudal thrombus in a concentration-dependent manner when the HPA concentration ranged from 82.2 µM to 328.6 µM.

The cardiac erythrocyte staining results are shown in Figure 4. Compared with the blank control group, AA significantly reduced the staining area and the staining intensity of zebrafish cardiac erythrocytes. After HPA exposure at concentrations of 164.3 and 328.6 µM, the area and intensity of cardiac erythrocyte staining significantly increased, which indicated strong antithrombotic activity. Combined with the results of the zebrafish caudal thrombus assay, the area and staining intensity of caudal thrombus was inversely proportional to the area and intensity of zebrafish caudal thrombus staining, which is consistent with previous research [24,25].

The number of migrating inflammatory cells in each group is shown in Figure 5. We counted the number of macrophages that migrated to the lateral line after the zebrafish cloaca, i.e., the number of green fluorescent dots in the red dashed area of Figure 5A. AA caused a significant increase in the number of migrating inflammatory cells. Like in the ASA group, at concentrations ranging from 82.2 to 657.2 µM, HPA markedly reduced the migration of inflammatory cells in zebrafish and reversed the inflammatory response caused by AA.

We analyzed the blood flow velocity and counted the heart rate in one minute for each group of zebrafish (Figure 6). Zebrafish in the AA group showed a massive reduction in blood flow velocity and heart rate. Compared with those in the AA group, the blood flow velocity and heart rate of zebrafish in the HPA groups exposed to 164.3, 328.6, and 657.2 µM were clearly greater than those in the AA group. However, zebrafish blood flow velocity and heart rate were lower at an HPA concentration of 657.2 µM than at an HPA concentration of 164.3 and 328.6µM; this suggests an atypical dose effect for the action of HPA.

### 2.3. Functional Classification and Annotation of The Transcriptome

Figure 7 displays the distributions of the differentially expressed genes (DEGs). In the AA group, 5090 significant DEGs were found compared to those in the control group, comprising 2572 upregulated and 2518 downregulated genes. In contrast to the AA group, the HPA group exhibited 2641 differentially expressed genes (DEGs), comprising 1452 upregulated and 1189 downregulated genes. There were 1035 DEGs coexpressed in the two comparison combinations (AA vs. control and HPA vs. AA), of which 101 DEGs were upregulated in the combination of AA vs. control and downregulated in the combination of HPA vs. AA, and 90 DEGs were downregulated in the combination of AA vs. control and upregulated in the combination of HPA vs. AA. In brief, the expression of 191 DEGs showed opposite trends in the two comparisons above.

According to the Gene Ontology (GO) enrichment analysis of the 191 DEGs, the 30 most significant terms are shown in Figure 8. These significant DEGs were enriched in many gene functions, such as negative regulation of transcription, notochord development, trans-Golgi network transport vesicle membrane, RNA polymerase binding, and transcriptional activator activity.

The Kyoto Encyclopedia of Genes and Genomes (KEGG) enrichment analysis of the 191 DEGs revealed the 20 most significant pathways. As shown in Figure 9, these DEGs were primarily concentrated in pathways associated with metabolism, including biosynthesis of amino acids, glycine, serine and threonine metabolism, carbon metabolism, cysteine and methionine metabolism, glutathione metabolism, insulin signaling pathway, alanine, aspartate, glutamate metabolism, taurine and hypotaurine metabolism, one carbon pool by folate, 2-oxocarboxylic acid metabolism, lysine degradation, and arginine biosynthesis. Additionally, the apelin signaling pathway, aminoacyl-tRNA biosynthesis pathway, erbB signaling pathway, cofactor biosynthesis pathway, autophagy pathway, mTOR signaling pathway, notch signaling pathway, and adipocytokine signaling pathway were among the top 20 enriched factors.

### 2.4. qRT–PCR Analysis

To confirm the results of the transcriptome analysis, genes in the mTOR and autophagy signaling pathways were selected for qRT–PCR. As shown in Figure 10A, compared to those in the control group, AA significantly increased the expression of the *pik3r1*, *pik3r3b*, *akt1,* and *mtor* genes in the PI3K/AKT/mTOR signaling pathway. After incubation with HPA, the expression levels of these genes decreased. For the autophagy signaling pathway, the results showed (Figure 10B) that AA treatment downregulated the expression levels of *ulk1b*, *eif2ak3*, *becn1*, and *ambra1a* in zebrafish, while HPA treatment upregulated their expression levels. Therefore, the mechanism of HPA antithrombotic activity involved the downregulation of *pik3r1*, *pik3r3b*, *akt1,* and *mtor* and the upregulation of *ulk1b*, *eif2ak3*, *becn1*, and *ambra1a*.

## 3. Discussion

Thrombosis is a major cause of morbidity and mortality worldwide, and safe and reliable antithrombotic drugs are urgently needed [39,40]. Marine natural products are promising drug sources, and many natural products with antithrombotic activity have been isolated from marine sources [41,42]. In the zebrafish model, AA can be used to induce experimental thrombosis [43,44]. In this paper, AA-induced zebrafish of the AB lineage were used as a model to assess the antithrombotic effect of HPA. To determine the safe concentration of HPA for use in zebrafish for the evaluation of antithrombotic activity, we conducted an HPA safety concentration test and obtained an LC_1_ of 914.6 µM for HPA.

Thrombosis can prevent blood from returning to the heart and reduce the number of cardiac erythrocytes [45], which was consistent with the results of the AA-treated group, suggesting that AA induces thrombosis. HPA dramatically reduced the caudal thrombus area and restored the volume of blood returned to the heart. Thrombosis can slow blood flow [46] and lower the heart rate. In this study, we found that the blood flow velocity and heart rate were significantly greater in the HPA group than in the AA group. Thrombosis can trigger an immune response, which in turn is accompanied by inflammation [47]. Our experiments also indicated that the number of migrating inflammatory cells significantly increased in the AA-treated group and then obviously decreased in the HPA-treated group. The results of the bioassay protocols indicated an atypical dose effect for the antithrombotic activity of HPA. Based on the results of the caudal thrombus assay, we can confirm that HPA with a concentration of 82.2–328.6 µM has antithrombotic activity in a dose-dependent manner. In the other assay, we found that when the concentration of HPA was 82.2 µM, there was no significant difference in the cardiac erythrocyte, caudal blood flow velocity, or heart rate assay compared with that in the AA group. This may be due to the limited antithrombotic effect of HPA at 82.2 µM, which leads to the insignificance of these indicators. In addition, the antithrombotic activity was lower in the 657.2 µM HPA-treated group than in the 328.6 µM HPA-treated group, which may be due to excessive autophagy. In contrast to appropriately enhanced autophagy, excessive autophagy can cause cell death [48], while macrophage autophagic death releases inflammatory factors and triggers the inflammatory response [49], and autophagic death of smooth muscle cells and vascular endothelial cells reduces plaque stability and promotes thrombosis [50].

According to transcriptome analysis and qRT–PCR validation, HPA exerts its antithrombotic effects by inhibiting the PI3K/AKT/mTOR signaling pathway and enhancing autophagy. Previous studies have shown that platelet activation and aggregation are the pathological basis of thrombosis and that the rupture or erosion of atherosclerotic plaques triggers platelet aggregation, leading to local coagulation activation and thrombosis [51,52]. The PI3K/AKT/mTOR signaling pathway plays an important role in the thrombosis process [53]. Inhibition of the PI3K/AKT pathway can depress platelet activation and adhesion, thereby suppressing thrombosis [54,55,56], and mTOR is phosphorylated at Ser^2448^ by the PI3K/Akt signaling pathway and plays a key role in platelet aggregation [57]. In this study, HPA significantly downregulated the mRNA expression levels of *pik3r1*, *pik3r3b*, *akt1,* and *mtor*, suggesting that HPA inhibits the PI3K/AKT/mTOR signaling pathway. In addition, the inhibition of the PI3K/AKT/mTOR pathway can induce autophagy [58], and appropriately enhancing autophagy in macrophages can suppress the formation and development of atherosclerotic plaques, promote plaque stabilization, and prevent plaque rupture, thus inhibiting thrombosis [7,52,59,60]. The *ulk1b* gene plays an important role in autophagy and as downstream genes of *ulk1b*, *becn1,* and *ambra1a* can participate in the process of autophagic membrane segregation [61,62], and *eif2ak3* can induce autophagy [63]. In our study, the mRNA expression levels of *ulk1b*, *eif2ak3*, *becn1,* and *ambra1a* were upregulated by 164.3 and 328.6 µM HPA, suggesting that HPA can induce autophagy.

## 4. Experimental Methods

### 4.1. General Experimental Procedures

Images were observed and captured with an SZX16 fluorescence microscope, a DP2-BSW image acquisition system (Olympus, Tokyo, Japan), and an AXIO-V16 fluorescence microscope (Zeiss, Oberkochen, Germany). An HPG280-BX Illumination Incubator (Donglian Electronic Technology Development Co., Ltd., Harbin, China) was used for zebrafish culture after drug administration. A zebrafish culture system (ESEN Technology Development Co., Ltd., Beijing, China) was used to cultivate the zebrafish. NMR spectra were recorded on a Bruker Avance spectrometer (Bruker, Billerica, MA, USA) operating at 400 (^1^H) and 100 (^13^C) MHz with TMS as an internal standard. ESI-MS data were acquired on an Agilent 6210 ESI/TOF mass spectrometer (Agilent, Santa Clara, CA, USA).

### 4.2. Fungal Material

After isolating the fungus from a sea water sample taken in 2013 from the Indian Ocean (88°59′51″ E, 2°59′54″ S), we identified it as *Emericellopsis maritima* (GenBank: MH871998.1) through ITS sequence analysis and rDNA amplification. The strain was placed at the Institute of Biology Institute of the Shandong Academy of Sciences’ Drug Screening Research Laboratory.

### 4.3. Fermentation, Extraction, and Isolation

The fermentation media consisted of 50 g of rice and 75 mL of sea water in 500 mL Erlenmeyer flasks, and the strains were cultured in 40 bottles of fermentation media for 60 days at room temperature. All of the fermented material was extracted twice with ethyl acetate and a mixture of dichloromethane:methanol (1:1). Then, the extract was concentrated under reduced pressure, dispersed with water, and extracted with an equal volume of ethyl acetate 4 times to obtain the ethyl acetate phase, which was further concentrated under reduced pressure to obtain the crude extract. Through silica gel column chromatography (CC) and elution with petroleum ether:ethyl acetate (100:0–0:100), the crude extract was separated into 7 fractions, Frs. 1–7. Fr. 6 was subjected to silica CC and then purified using HPLC using 30% MeOH–H_2_O to give the compound HPA.

Compound HPA: White crystal powder, ESI-MS *m/z*: 151 [M – H]^–^. ^13^C NMR (100 MHZ, CD_3_OD) δ_C_: 175.0 (C-8), 156.0 (C-4), 129.9 (C-2, 6), 125.6 (C-1), 114.8 (C-3, 5), 39.9 (C-7); ^1^H NMR (400 MHZ, CD_3_OD) δ_H_: 7.09 (2H, d, J = 8.8 Hz, H-2, 6), 6.72 (2H, d, J = 8.8 Hz, H-3, 5), 3.47 (2H, s, H-7). The above data are consistent with literature reports [64].

### 4.4. Safety Concentration Test

We selected healthy zebrafish larvae at 72 hpf under a stereoscopic microscope, transferred them to 24-well plates (ten larvae per well), and treated them for 6 h with different concentrations of HPA (0, 164.3, 328.6, 657.2, 1314.5, 1971.7, 2629.0, 3286.2, 3943.5, or 5258.0 µM). All treatments were performed in triplicate, and the number of subjects per condition was 30. Zebrafish were observed and imaged on a camera using an Olympus microscope, and the mortality rate of the zebrafish in each group was determined. The lethal curve was plotted using GraphPad Prism 9.0 (GraphPad Software, Inc., San Diego, CA, USA) with the HPA concentration and mortality rate as the horizontal and vertical coordinates, respectively, and then the LC_1_ was calculated through Nonlinear regression (curve fit).

### 4.5. Bioassay Protocols

#### 4.5.1. Zebrafish Maintenance

The zebrafish (*Danio rerio*) strains used in this experiment were the AB wild-type and *Tg (zlyz:EGFP)* transgenic lines. Male and female zebrafish were maintained separately at 28.0 °C ± 0.5 °C in an automatic circulating aquarium with a 14/10 h light/dark photoperiod. The embryos were obtained from natural spawning. After fertilization, the eggs were collected, cleaned, and sterilized with methylene blue solution and then placed in zebrafish embryo culture water supplemented with 5.0 mM of NaCl, 0.17 mM of KCl, 0.4 mM of CaCl_2_, and 0.16 mM of MgSO_4_ in a constant temperature light incubator at 28.0 °C ± 0.5 °C. The culture water was changed every 24 h.

#### 4.5.2. Chemical Treatment

Healthy zebrafish larvae were chosen at 72 h post-fertilization (hpf) and placed into 24-well plates (ten larvae per well). The plants were randomly divided into 7 groups: a blank control group (fresh fish farming water), a thrombus model group (80 µM AA), a positive control group (80 µM AA + 125.0 µM ASA), and groups treated with different concentrations of HPA (80 µM AA + 82.2, 164.3, 328.6, or 657.2 µM HPA). After treatment with ASA and HPA for 6 h, the solution was aspirated, then the blank control group was treated with fish farming water, and all other groups were treated with 80 µM AA. The plates were incubated at 28 ± 0.5 °C for 1 h. All treatments were performed in triplicate, and the number of subjects per condition was 30.

#### 4.5.3. Caudal Thrombus Assay

At the end of drug treatment, the solution was pipetted out, and then the zebrafish larvae were stained with 1.0 mg/mL of O-dianisidine dye liquor for 10 min in the dark. After washing with zebrafish culture solution three times, each zebrafish was fixed with 4% paraformaldehyde, and then the caudal thrombi of the zebrafish were observed and photographed under a Zeiss fluorescence microscope. The stained area and intensity of the caudal thrombus in the zebrafish were measured and calculated using Image-Pro Plus 5.0 software (Media Cybernetics, Inc., Bethesda, MD, USA).

#### 4.5.4. Cardiac Erythrocyte Assay

After staining, rinsing, and fixation (as described in Section 4.5.3), each zebrafish was observed and photographed under a Zeiss fluorescence microscope. The thrombus density was quantified according to the stained area, and the intensity of erythrocytes in the heart was analyzed using Image-Pro Plus 5.0 software.

#### 4.5.5. Caudal Blood Flow Velocity and Heart Rate Assay

After drug treatment, each group of zebrafish was washed three times with zebrafish culture water and then observed with an Olympus fluorescence inverted microscope. The heart rates of the zebrafish were obtained, and a video of blood flow for 15 s was recorded from the postcloacal vessels of the zebrafish using the blood flow system Zebralab (ViewPoint, Lyon, France). Analysis of blood flow velocity was then performed using MicroZebraLab BloodFlow 3.4.6 software (ViewPoint, Lyon, France).

#### 4.5.6. Inflammatory Cell Assay

The transgenic zebrafish *Tg (zlyz:EGFP)* with green fluorescent protein (EGFP)-labeled macrophages was selected for this experiment. After drug treatment, images of the caudal inflammatory cells in the zebrafish were recorded under an Olympus inverted fluorescence microscope. The number of caudal inflammatory cells that migrated to the tail notochord and above was counted.

### 4.6. Transcriptome Analysis

At 72 hpf, normal zebrafish larvae were selected and transferred to 6-well plates. The blank control group (fresh fish farming water), thrombus model group (80 µM AA), and HPA group (80 µM AA + 164.3 or 328.6 µM HPA) were randomly divided, with 50 larvae in each group. Drug treatment for each group followed the same strategy as described in Section 4.5.2. All treatments were carried out in triplicate. TRIzol reagent was used to extract total RNA from the zebrafish. An Agilent 2100 Bioanalyzer (Agilent Technologies, Santa Clara, CA, USA) was used to assess the integrity of the RNA. The raw material for the RNA sample preparations was total RNA. Briefly, poly-T oligo-attached magnetic beads were used to separate mRNA from the total RNA. Divalent cations were used to carry out fragmentation in First Strand Synthesis Reaction Buffer (5X) at a high temperature, and then the libraries were established using the NEB common library building method [65]. Next, Novogene Co., Ltd. (Beijing, China) carried out transcriptome sequencing and analysis.

DESeq2 (version 1.20.0) software (Novogene Co., Ltd., Tianjin, China) was used to carry out differential expression analyses between the two comparison combinations (AA vs. control and HPA vs. AA). A model based on the negative binomial distribution was used to identify DEGs in the gene expression data. P values less than 0.05 indicated significantly differentially expressed genes, which were chosen for further examination. Gene Ontology (GO) and Kyoto Encyclopedia of Genes and Genomes (KEGG) pathway analyses were used to examine the statistical enrichment of DEGs.

### 4.7. qRT–PCR Assay

Based on the results of the transcriptome analysis, 7 key DEGs were selected for further qRT–PCR analyses. The names and sequences of the primers used are shown in Table 1. Sample treatment and collection were the same as those described in Section 4.6, and total RNA was extracted from the samples with an RNA extraction kit (Vazyme, Nanjing, China). cDNA was obtained through reverse transcription of total RNA with a reverse transcription kit (Vazyme, Nanjing, China). The qRT–PCR system for each well in the 8-tube strip consisted of 2 µL of cDNA (10 ng/µL), 0.4 µL of forward primer, 0.4 µL of reverse primer, 7.2 µL of ddH_2_O, and 10 µL of SYBR qPCR master mix (Vazyme, Nanjing, China). qRT–PCR was performed in a LightCyler 96 system (Roche, Basel, Switzerland), and the reaction conditions were 30 s at 95 °C, followed by 40 cycles of 95 °C for 15 s and 60 °C for 30 s. *β-actin* expression levels were used as a normalization control for the expression of other genes. The qRT–PCR data were analyzed using the comparative threshold cycle method (2^–ΔΔCt^) to calculate the relative expression of each group of genes.

### 4.8. Statistical Analysis

Software called GraphPad Prism 9.0 was used to process the statistical analysis. All of the experimental data are shown as mean ± SEM. The comparison between groups was performed using One-Way ANOVA and Dunnett’s *t*-test. At *p* values less than 0.05, the differences between the groups were considered significant.

## 5. Conclusions

To summarize, 4-hydroxyphenylacetic acid was isolated from the marine fungus *Emericellopsis maritima* Y39–2, and its antithrombotic activity was determined using a zebrafish model. HPA significantly improved AA-induced thrombus in zebrafish. Specifically, it can increase cardiac erythrocytes, blood flow velocity, and heart rate in zebrafish, reduce caudal thrombus, and reverse the inflammatory response caused by AA. Further research showed that HPA exerted its antithrombotic effect by inhibiting the PI3K/AKT/mTOR signaling pathway and inducing autophagy.

## Figures and Tables

**Figure 1 marinedrugs-22-00148-f001:**
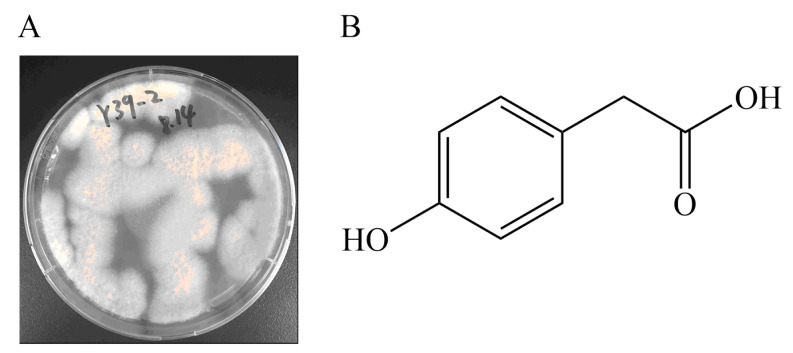
(**A**) The marine-derived fungus *Emericellopsis maritima* Y39–2. (**B**) Structure of the HPA.

**Figure 2 marinedrugs-22-00148-f002:**
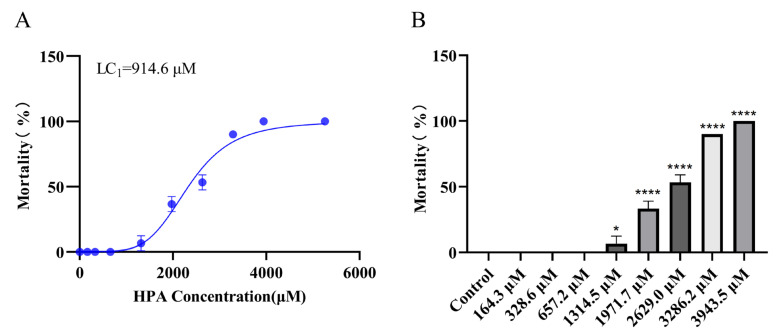
Observation of 72 hpf zebrafish larvae exposed to HPA for 6 h. (**A**) Mortality curve and (**B**) mortality rate at 6 hpe. The data are presented as the means ± SEMs; **** *p* < 0.0001 and * *p* < 0.05 compared to the control group.

**Figure 3 marinedrugs-22-00148-f003:**
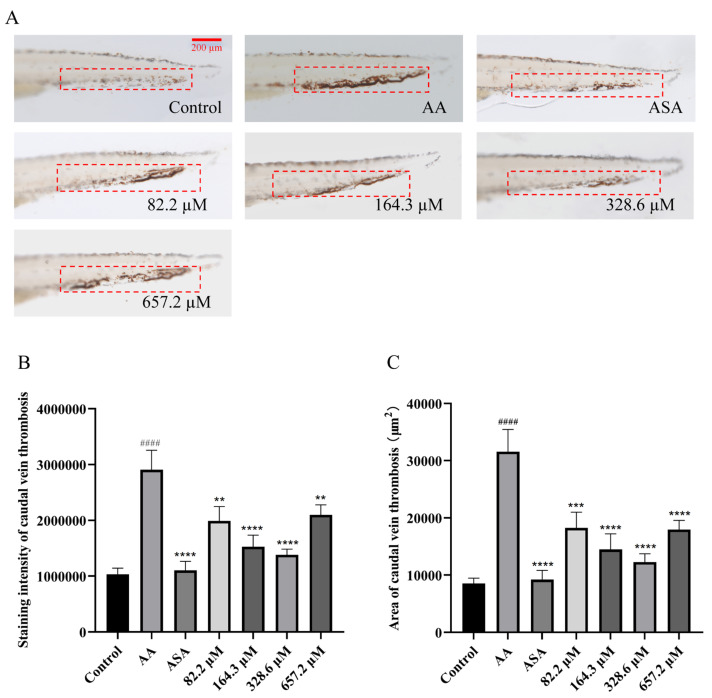
Reduction of caudal thrombus in zebrafish treated with HPA. (**A**) Typical images of caudal thrombus staining in zebrafish. Quantitative analysis of (**B**) the intensity and (**C**) the area of caudal thrombus staining. The data are presented as the means ± SEMs; ^####^
*p* < 0.0001 compared to the control group; **** *p* < 0.0001, *** *p* < 0.001, and ** *p* < 0.01 compared to the AA group.

**Figure 4 marinedrugs-22-00148-f004:**
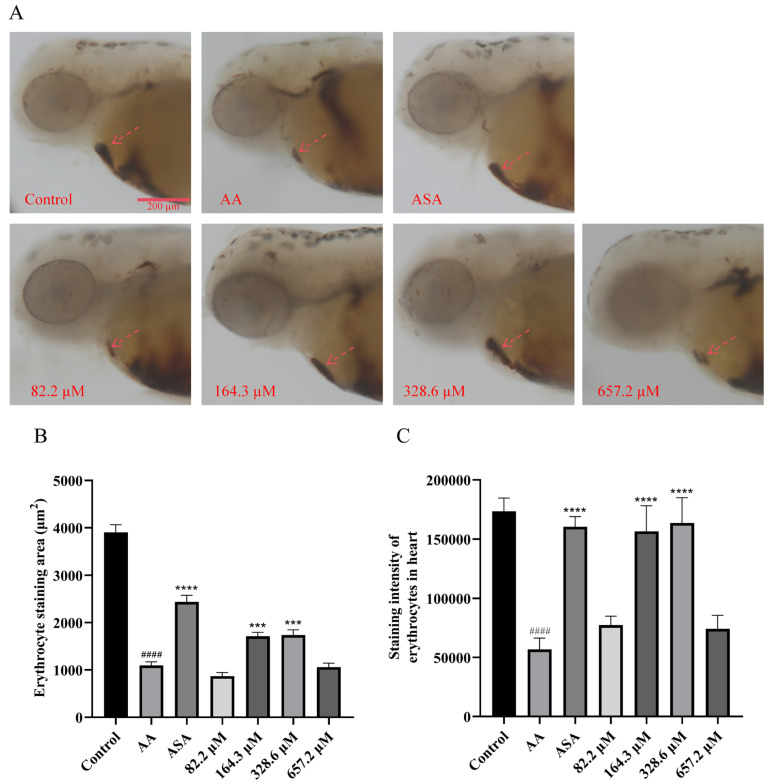
Increase in cardiac erythrocytes in zebrafish treated with HPA. (**A**) Typical images of erythrocyte staining in zebrafish heart (shaded areas are indicated by arrows). Quantitative analysis of (**B**) the area and (**C**) the intensity of cardiac erythrocyte staining. The data are presented as the means ± SEMs; ^####^
*p* < 0.0001 compared to the control group; **** *p* < 0.0001 and *** *p* < 0.001 compared to the AA group.

**Figure 5 marinedrugs-22-00148-f005:**
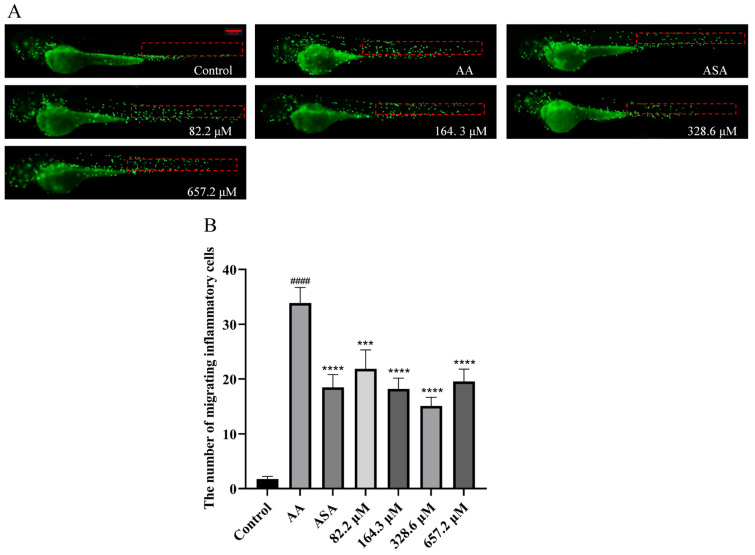
Reduction in the AA-induced inflammatory response in zebrafish treated with HPA. (**A**) Typical images of inflammatory cell migration in zebrafish from all groups. (**B**) Quantitative evaluation of the quantity of migrating inflammatory cells. The data are presented as the means ± SEMs; ^####^
*p* < 0.0001 compared to the control group; **** *p* < 0.0001 and *** *p* < 0.001 compared to the AA group.

**Figure 6 marinedrugs-22-00148-f006:**
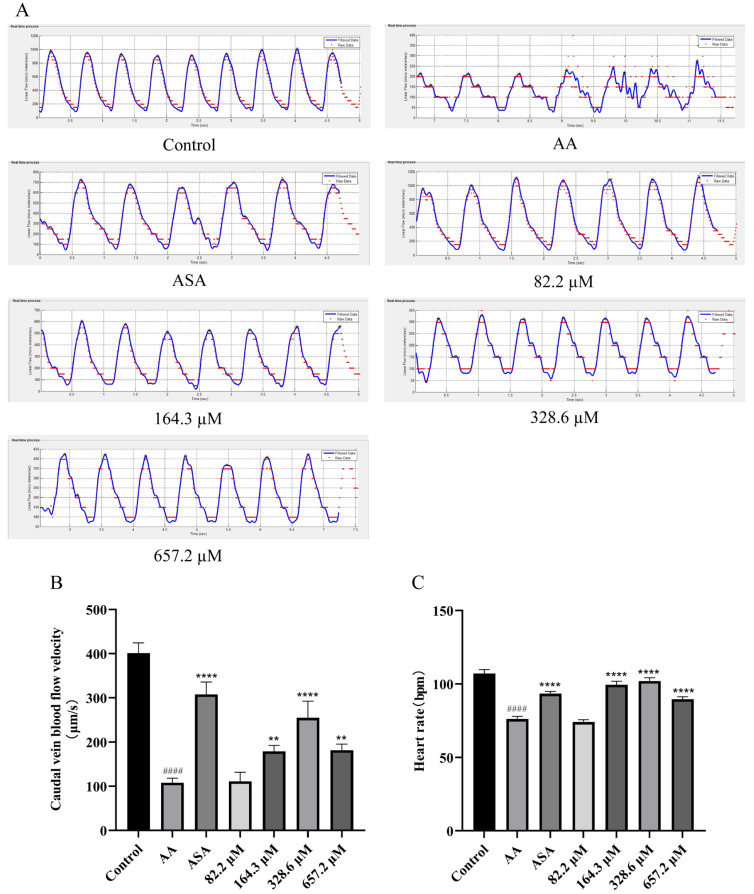
Improvement in caudal blood flow velocity and heart rate in zebrafish treated with HPA. (**A**) Blood flow dynamics of the caudal vein in zebrafish. Quantitative analysis of (**B**) the caudal blood flow velocity and (**C**) the heart rate in one minute. The data are presented as the means ± SEMs; ^####^
*p* < 0.0001 compared to the control group; **** *p* < 0.0001 and ** *p* < 0.01 compared to the AA group.

**Figure 7 marinedrugs-22-00148-f007:**
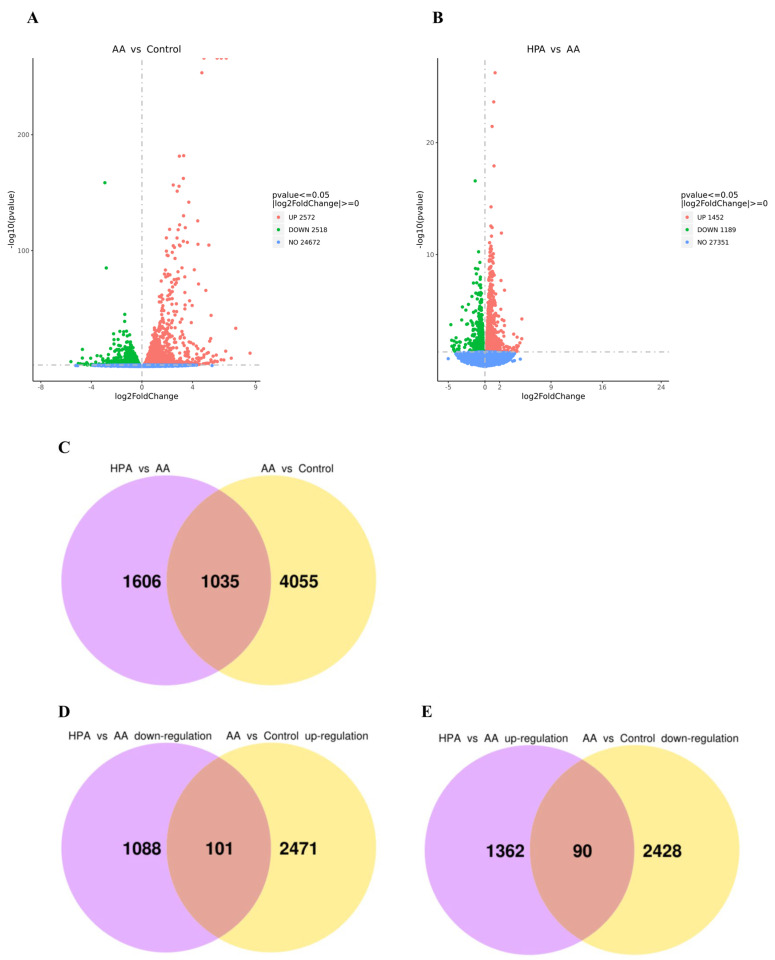
Differentially expressed genes (DEGs) in zebrafish. (**A**,**B**) Volcano plots for DEGs of AA vs. control and HPA vs. AA. (**C**) Venn plot for the DEGs of AA vs. control and HPA vs. AA. (**D**) Venn plot of the DEGs downregulated in the HPA group vs. the AA group and upregulated in the AA group vs. the control group. (**E**) Venn plot of the DEGs upregulated in the HPA group vs. the AA group and downregulated in the AA group vs. the control group.

**Figure 8 marinedrugs-22-00148-f008:**
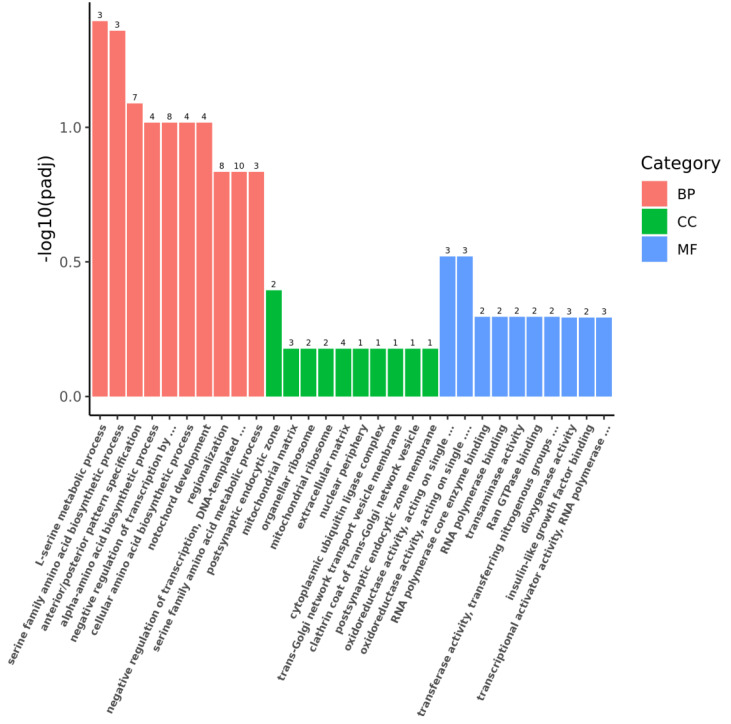
GO enrichment analysis results of 191 DEGs obtained from screening. BP in red: biological processes, CC in green: cellular composition, MF in blue: molecular function.

**Figure 9 marinedrugs-22-00148-f009:**
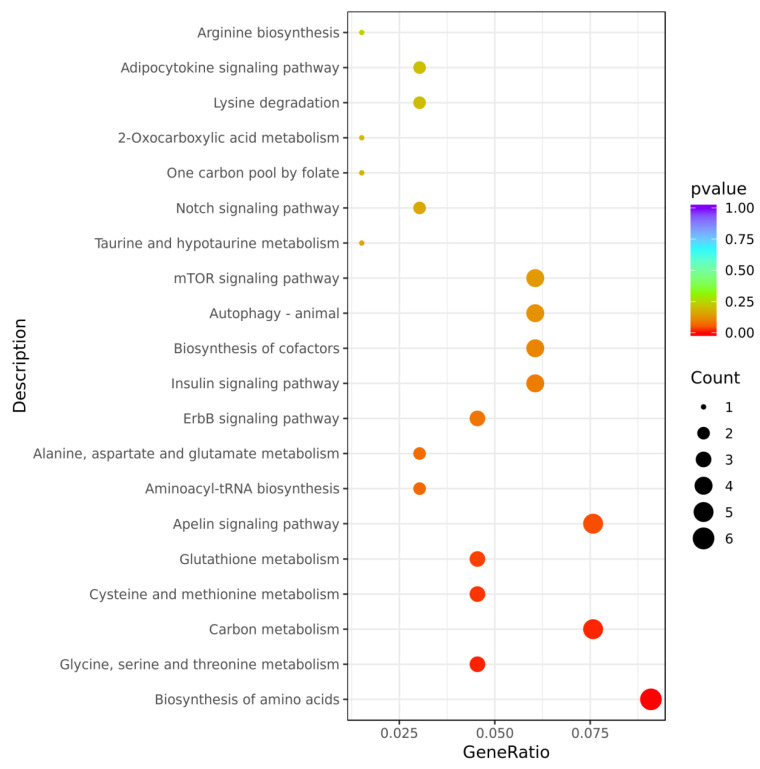
KEGG enrichment analysis results of 191 DEGs obtained from screening.

**Figure 10 marinedrugs-22-00148-f010:**
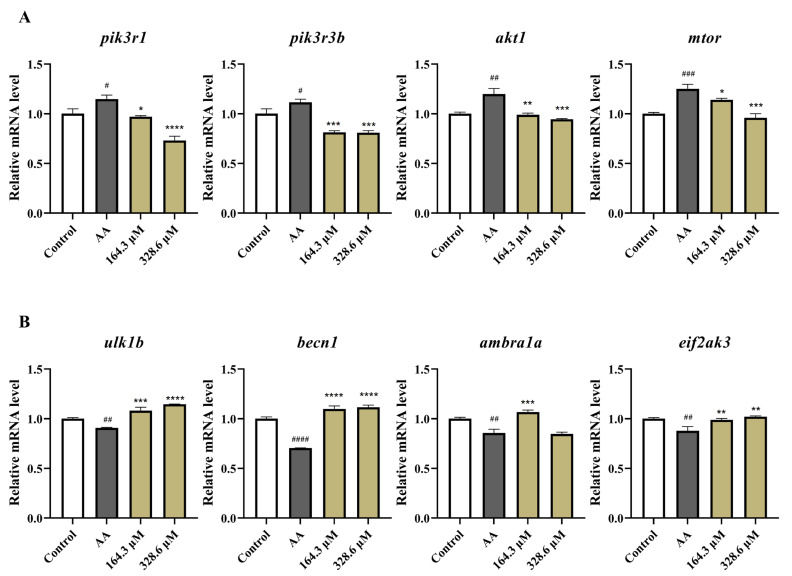
The results of qRT–PCR analysis. Quantitative analysis of the mRNA expression levels of genes associated with (**A**) the PI3K/AKT/mTOR signaling pathway and (**B**) the autophagy signaling pathway. The data are presented as the means ± SEMs; **^####^** *p* < 0.0001, **^###^ ***p* < 0.001, **^##^ ***p* < 0.01, and **^#^ ***p* < 0.05 compared to the control group; **** *p* < 0.0001, *** *p* < 0.001, ** *p* < 0.01, and * *p* < 0.05 compared to the AA group.

**Table 1 marinedrugs-22-00148-t001:** The sequences of primers.

Gene Name	Forward Primer (5′→3′)	Reverse Primer (5′→3′)
*β-actin*	ACCACGGCCGAAAGAGAAAT	GATACCGCAAGATTCCATACCC
*pik3r1*	TTCAGACATCAGCCCACAAGTT	GCCTGACTGTCACTCCATTCG
*pik3r3b*	TGACACCAGTAAACGGCAATG	TGTTTCCACCTTTCCTTAGCG
*akt1*	GTGAAGGAGAAAGCAACAGGCA	ATGACTTCAGAGCCGTCAGGAA
*mtor*	ACGCTGCACGCACTGATTC	AGAGTCAAATGGTCATAGTCAGGG
*ulk1b*	GTGCCTTCGCAGTGGTTTT	TGCTGTAGGAATACTCGGATGGT
*eif2ak3*	AACCGTCTGGAGACACCGAT	TGCTGAGGCTTGAGGATACCA
*becn1*	AGATGGCGTGGCTCGAAAAT	GCACTCCTCACAAAGTGGGT
*ambra1a*	ACGCACTCATCCGTCAATAGG	TTCACCGACGCATTACTGATTTC

## Data Availability

The data presented in this study are available on request from the corresponding author.

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
