# Peer review of "Marine-Fungus-Derived Natural Compound 4-Hydroxyphenylacetic Acid Induces Autophagy to Exert Antithrombotic Effects in Zebrafish"

_marinedrugs, 2024, doi:10.3390/md22040148_

Round 1

Reviewer 1 Report

Comments and Suggestions for Authors

The data are interesting, correctly presented graphically, but discussed in a superficial form in the text. For example, Fig3A-B-C is discussed very briefly (lines 104-108), without mentioning the atypical dose effect. The dose-dependence of the erythrocyte staining effect is a little strange, being more effective at 12.5 and 100 ug/ml compared to 25-50 ug/ml. A more complete presentation of all the data is important. In general, the lack of a clear dose-dependent effect is a problem with this work. In each case (Fig 3-6), the response to the HPA treatment looks a little erratic, with no concentration-dependence (particularly obvious in Fig. 6). This fact must be discussed further.

The manuscript is perfectible. However, it is an interesting and innovative work, worthy of publication in the journal.

Other points

-   Define AA (arachidonic acid) in the abstract

-   lines 48-49: Emericellopsis has been found to comprise… à Emericellopsis comprises….

-   Lines 70-71: Previously, the marine-derived fungus Emericellopsis maritima Y39-2 was used as a zebrafish model…. A reference to this previous work should be added.

-   Fig 1 is a little short and minimalist. I would suggest to include also an illustration of the fungus Emericellopsis maritima Y39-2, if possible.

-   The 8 photos of larva in Fig2A look similar. Difference, if any, or specific morphology changes should be discussed. Otherwise this Fig2A may be removed or simplified (no need to show 8 identical images).

-   Figs 4A and 5A: change “aspinin” -> Aspirin. Throughout the manuscript, it would be more appropriate to use the scientific name acetylsalicylic acid (ASA).

Comments on the Quality of English Language

OK for me.

Author Response

Dear reviewer,

Thank you very much for all your kind review on our manuscript and for your valuable suggestions.

The comments from you have been concerned in details. According to those helpful comments, we made a careful revision on the original manuscript “Marine fungus-derived natural compound 4-hydroxyphenylacetic acid induces autophagy to exert antithrombotic effects in zebrafish” (Manuscript ID: marinedrugs-2912157). Here we submit you the revised version of the manuscript.

We have managed to minimize all kinds of errors and obscurities in order to make it understood without difficulty. We hope it is not about to disappoint you and will feel very grateful if it is able to fulfill the standards of your journal and be accepted. Anyway, a lot of appreciation will be paid to you for your helpful work, beneficial suggestions and extraordinary patience during this process.

All the changes were listed as follows:

Remark 1:

Comments to the Author

The data are interesting, correctly presented graphically, but discussed in a superficial form in the text. For example, Fig3A-B-C is discussed very briefly (lines 104-108), without mentioning the atypical dose effect.

Answer:

In lines 125-128, 135-137, 152-156. we have provided a more detailed explanation and discussion of the experimental results.

In lines 110 and 114, we have explained that the observed and statistically analyzed data are the tail thrombus area and tail thrombus staining intensity.

In lines 125-128, we have explained that combined with the results of the zebrafish caudal thrombus assay, the area and staining intensity of caudal thrombu was inversely proportional to the area and intensity of zebrafish caudal thrombus staining, which is consistent with previous research (Howe, K.; et, al. The zebrafish reference genome sequence and its relationship to the human genome. Nature 2013, 496, 498-503.)(Xu, N.; et, al. Screening of antithrombotic active Fractions and chemical constituents in Sparganii Rhizoma based on zebrafish model. Chin. J. Hosp. Pharm. 2019, 39, 1439-1443.).

In lines 135-137, we noted that the inflammatory cells were counted were macrophages that migrated to the zebrafish lateral line after the zebra cloaca.

In lines 152-156, we have explained that an atypical dose effect for the action of HPA.

The results of the bioassay protocols indicated an atypical dose effect for the antithrombotic activity of HPA. Based on the results of caudal thrombus assay, we can confirm that HPA with a concentration of 82.2-328.6 µM has antithrombotic activity in a dose-dependent manner. In other assay, we found that when the concentration of HPA was 82.2 µM, there was no significant difference in the cardiac erythrocyte, caudal blood flow velocity and heart rate assay compared with that in AA group. This may be due to the limited antithrombotic effect of HPA at 82.2 µM, which leads to the insignificance of these indicators. In addition, the antithrombotic activity was lower in the 657.2 µM HPA-treated group than in the 328.6 µM HPA-treated group, which may be due to excessive autophagy. In contrast to appropriately enhanced autophagy, excessive autophagy can cause cell death (Schrijvers, D.M.; et, al. Autophagy in atherosclerosis: a potential drug target for plaque stabilization. Arterioscler., Thromb., Vasc. Biol. 2011, 31, 2787-2791.), while macrophage autophagic death releases inflammatory factors and triggers the inflammatory response (Denton, D.; et, al. Autophagy as a pro-death pathway. Immunol. Cell Biol. 2015, 93, 35-42.), and autophagic death of smooth muscle cells and vascular endothelial cells reduces plaque stability and promotes thrombosis (Martinet, W.; et, al. Autophagy in atherosclerosis: a cell survival and death phenomenon with therapeutic potential. Circ. Res. 2009, 104, 304-317.), this was explained in lines 238-251.

Remark 2:

Comments to the Author

The dose-dependence of the erythrocyte staining effect is a little strange, being more effective at 12.5 and 100 µg/mL compared to 25-50 µg/mL.

Answer:

When thrombus developed in zebrafish, the amount of blood returned to the heart will be reduced (Zhu, H.; et, al. Wuliangye Baijiu but not ethanol reduces cardiovascular disease risks in a zebrafish thrombosis model. NPJ Sci. Food 2022, 6, 55.), which is shown in the experiment as a decrease in the area and intensity of red blood cell staining in the heart. The larger the area and intensity of red blood cell staining in the heart, the less thrombus in the body of zebrafish and the better the antithrombotic activity of the drug. Therefore, the antithrombotic effect of HPA at concentrations of 82.2 µM (12.5 µg/mL) and 657.2 µM (100 µg/mL) is weaker than that of HPA at 164.3-328.6 µM (25-50 µg/mL).

Remark 3:

Comments to the Author

A more complete presentation of all the data is important. In general, the lack of a clear dose-dependent effect is a problem with this work. In each case (Fig 3-6), the response to the HPA treatment looks a little erratic, with no concentration-dependence (particularly obvious in Fig. 6). This fact must be discussed further.

Answer: As for the problem of concentration dependence, we have further explained and discussed it in line 152-156 and section “3. Discussion” lines 238-251, some of the activity indexes were not significantly different at HPA concentration of 82.2 µM (12.5 µg/mL) compared with the AA group, and the antithrombotic activity at HPA concentration of 657.2 µM (100 µg/mL) was lower than that at HPA concentration of 328.6 µM (50 µg/mL), which suggests that there is an atypical dose effect of the antithrombotic activity of HPA, which may be due to the weak antithrombotic activity of HPA at HPA concentration of 82.2 µM (12.5 µg/mL), which was triggered excessive autophagy by the HPA concentration of 657.2 µM (100 µg/mL).

Other points

Point 1

 - Define AA (arachidonic acid) in the abstract.

In line 28, page 1, it has been added.

Point 2

-lines 48-49: Emericellopsis has been found to comprise…à Emericellopsis comprises….

-Lines 70-71: Previously, the marine-derived fungus Emericellopsis maritima Y39-2 was used as a zebrafish model…. A reference to this previous work should be added.

In lines 71-72, We have outlined our previous work, including small-scale fermentation, extraction and testing for antithrombotic activity.

Point 3

-Fig 1 is a little short and minimalist. I would suggest to include also an illustration of the fungus Emericellopsis maritima Y39-2, if possible.

In Figure 1, We have added illustration of the fungus Emericellopsis maritima Y39-2 as suggested.

Point 4

-The 8 photos of larva in Fig2A look similar. Difference, if any, or specific morphology changes should be discussed. Otherwise this Fig2A may be removed or simplified (no need to show 8 identical images).

In Figure 2, We have deleted the photos of zebrafish larva.

Point 5

-Figs 4A and 5A: change “aspinin” -> Aspirin. Throughout the manuscript, it would be more appropriate to use the scientific name acetylsalicylic acid (ASA).

According your suggestion, we have used the scientific name acetylsalicylic acid (ASA) in this article. See Figures 3- 6, lines 40, 112, 139, 330, 332.

Thank you very much again for your patience and consideration.

Yours sincerely,

Xiaobin Li

Reviewer 2 Report

Comments and Suggestions for Authors

Xin et al. present a prospective investigation of the compound 4-hydroxyphenylacetic acid as an antithrombotic agent via induced autophagia in a zebrafish model. Using a variety of molecular methods, the authors determined the concentration of HPA that was nonlethal and most likely to diminish what they posited was clot formation in the tail. Lastly, using a variety of gene identification methods, the mechanism of the phenomena was purportedly determined. I have a number of comments.

Please express the concentrations of HPA in micromolar.

Figure 2. How was mortality determined? Please indicate the number of larvae were observed per condition.

Please make clear what the experimental conditions are. The data in results should indicate that AA was present when different concentrations of HPA were added. Also, indicate the order of addition – HPA first, then AA; or, AA first, then HPA.

Frankly, a brief description of the model is indicated. The reader should not have to sift through the other sections of the paper to understand the paradigm presented in terms of where the red cells go during thrombus formation. Also, it is not made clear if what we are seeing is a thrombus in the tail or just redistribution of red cells based on slower flow secondary to vasoconstriction by AA. The authors need to make a better case for the model in Introduction with some better detail in Results.

Figures 3 and 4. These figures should have the order switched, with figure 4 presented first as a new figure 3. It was confusing to first be presented with the erythrocytes in the heart as a measure of antithrombotic activity. If the increase and decrease in thrombus formation in the tail is presented first, then it will be clearer what the observations of the heart mean. Aspirin is also misspelled in the “A” panel of figure 4. Lastly, indicate the number of subjects per condition.

Figure 5. There is not enough detail in the text or figure legend. What sort of inflammatory cells are we seeing? What is the technique/method used? Immunofluorescence? Again, number of larvae per condition?

Figure 7 is unreadable. Please break it up into at least two figures so the text within the figure can be read.

Figure 10. As with figure 7, this figure is unreadable. Please modify/split up the figure.

In Methods, how did the authors make certain that they had produced HPA (section 4.3)? The method is clear, but were there standards used to verify the compound purified? Was mass spec performed? What was the purity?

Section 4.4. So, were three replicates of conditions with 10 larvae per condition the model used? Also, what criteria were used to decide if a fish was healthy or not? What criteria were used to define death?

Please indicate the vendors for the hardware and software used to conduct these experiments. Please indicate the dyes used and concentration in the immunofluorescence assays. Define the characteristics of the inflammatory cells documented. As written, these experiments cannot be repeated.

There is no comprehensive statistical section to present the methods, software or power of the various analyses found in the figures. The survival curve was “calculated.” Software and rationale for methods chosen must be presented.

In summary, the authors presented a series of experiments that seem to support the antithrombotic effect of HPA in zebrafish. However, serious issues must be resolved to make certain the conclusions drawn are supported by the data presented.

Author Response

Dear reviewer,

Thank you very much for all your kind review on our manuscript and for your valuable suggestions.

The comments from you have been concerned in details. According to those helpful comments, we made a careful revision on the original manuscript “Marine fungus-derived natural compound 4-hydroxyphenylacetic acid induces autophagy to exert antithrombotic effects in zebrafish” (Manuscript ID: marinedrugs-2912157). Here we submit you the revised version of the manuscript.

We have managed to minimize all kinds of errors and obscurities in order to make it understood without difficulty. We hope it is not about to disappoint you and will feel very grateful if it is able to fulfill the standards of your journal and be accepted. Anyway, a lot of appreciation will be paid to you for your helpful work, beneficial suggestions and extraordinary patience during this process.

All the changes were listed as follows:

Remark 1:

Comments to the Author

 Please express the concentrations of HPA in micromolar.

Answer:

 According to your suggestion, we have expressed the concentration of HPA with micromolar in Figures 2-6 and Fig 10, lines 97, 98, 101, 103, 115, 124, 139, 151-152, 227, 240, 242, 244, 246, 309-310, 331, 363.

Remark 2:

Comments to the Author

 Figure 2. How was mortality determined? Please indicate the number of larvae were observed per condition.

Answer:

We determined the mortality rate by comparing the number of deaths per well of a 24-well plate to the total number of fish per well (ten larvae per well), Three replicates were performed for each concentration, resulting in a total of 30 fish per concentration. The number of juvenile fish observed under different conditions is shown in the table below.

HPA concentrations

(µM)

Mortality (%)

Number of zebrafish alive

Well 1

Well 2

Well 3

Well 1

Well 2

Well 3

0

0

0

0

10

10

10

164.3

0

0

0

10

10

10

328.6

0

0

0

10

10

10

657.2

0

0

0

10

10

10

1314.5

0

10

10

10

9

9

1971.7

40

40

30

6

6

7

2629.0

50

50

60

5

5

4

3286.2

90

90

90

1

1

1

3943.5

100

100

100

0

0

0

5258.0

100

100

100

0

0

0

Remark 3:

Comments to the Author

 Please make clear what the experimental conditions are. The data in results should indicate that AA was present when different concentrations of HPA were added. Also, indicate the order of addition – HPA first, then AA; or, AA first, then HPA.

Answer:

In section “4.5.2. Chemical treatment” lines 327-333, we have explained that after treatment with ASA and HPA for 6 h, the solution was aspirated, then 80 µM AA was added to all groups except the blank control group. Specifically, the blank control and thrombus model groups treated zebrafish larvae with the zebrafish culture solution, the positive drug group treated zebrafish larvae with 22.5 µg/mL of Aspirin, and in the HPA group, the larvae were treated with 82.2 µM (12.5 µg/mL), 164.3µM (25 µg/ml), 328.6 µM (50 µg/mL) and 657.2 µM (100 µg/mL) HPA respectively, and all zebrafish were placed in a light incubator at 28.0 ℃±0.5 ℃, after 6 hours, the solution was aspirated, then the blank control group was treated with fish farming water, and the other groups were treated with 80 µM AA. The plates were incubated at 28 ± 0.5 ℃ for 1 h.

Remark 4:

Comments to the Author

Frankly, a brief description of the model is indicated. The reader should not have to sift through the other sections of the paper to understand the paradigm presented in terms of where the red cells go during thrombus formation. Also, it is not made clear if what we are seeing is a thrombus in the tail or just redistribution of red cells based on slower flow secondary to vasoconstriction by AA. The authors need to make a better case for the model in Introduction with some better detail in Results.

Answer:

Based on your suggestions, we have provided more details and arguments.

In page 3 lines 110 and 113,we illustrated that it was the caudal thrombus and its area that was observed and counted.

In lines 125-128, we have explained that combined with the results of the zebrafish caudal thrombus assay, the area and staining intensity of caudal thrombu was inversely proportional to the area and intensity of zebrafish caudal thrombus staining, which is consistent with previous research (Howe, K.; et, al. The zebrafish reference genome sequence and its relationship to the human genome. Nature 2013, 496, 498-503.)(Xu, N.; et, al. Screening of antithrombotic active Fractions and chemical constituents in Sparganii Rhizoma based on zebrafish model. Chin. J. Hosp. Pharm. 2019, 39, 1439-1443.).

In lines 135-137, we noted that the inflammatory cells were counted were macrophages that migrated to the zebrafish lateral line after the zebra cloaca.

In lines 152-156, we have explained that an atypical dose effect for the action of HPA.

Remark 5:

Comments to the Author

Figures 3 and 4. These figures should have the order switched, with figure 4 presented first as a new figure 3. It was confusing to first be presented with the erythrocytes in the heart as a measure of antithrombotic activity. If the increase and decrease in thrombus formation in the tail is presented first, then it will be clearer what the observations of the heart mean. Aspirin is also misspelled in the “A” panel of figure 4. Lastly, indicate the number of subjects per condition.

Answer:

Based on your suggestion, we have swapped the positions of Figures 3 and 4 and corrected spelling errors, and in the study, the number of subjects in per condition was 30 (lines 309-310 and 334-335).

Remark 6:

Comments to the Author

Figure 5. There is not enough detail in the text or figure legend. What sort of inflammatory cells are we seeing? What is the technique/method used? Immunofluorescence? Again, number of larvae per condition?

Answer:

In line 135-137, 357-358, we have explained that in the inflammatory cells assay, the inflammatory cells we saw labeled with green fluorescence were macrophages.

Transgenic zebrafish with macrophage labeled by green fluorescent protein was established by molecular cloning, microinjection and in vivo photography (Qin, S.; Chen, X.; Kong D.M. Construction of a transgenic zebrafish line consisting of green fluorescent labeled macrophages. J. Guizhou Univ. Chin. Med. 2011, 33, 133-135.)

The number of subjects in per condition was 30 (lines 334-335).

Remark 7:

Comments to the Author

Figure 7 is unreadable. Please break it up into at least two figures so the text within the figure can be read.

Answer:

At your suggestion, a cleaner version of Figure 7 has been adopted. Figures 7-A and 7-B represent the number and distribution of DEGs in the AA vs. Control comparison combination and the HPA vs. AA comparison combination, respectively; Figure 7-C expresses the genes co-expressed in the two comparison combinations; and Figure 7-D represents the DEGs that downregulated in the HPA group vs. the AA group and upregulated in the AA group vs. the control group. Figure 7-E represents the DEGs that upregulated in the HPA group vs. the AA group and downregulated in the AA group vs. the control group. In order to illustrate the process of target gene screening, so it is all represented on one graph.

Remark 8:

Comments to the Author

 Figure 10. As with figure 7, this figure is unreadable. Please modify/split up the figure.

Answer: 

In line 213 figure 10,we have modified Figure 10 to make it clearer.

Remark 9:

Comments to the Author

In Methods, how did the authors make certain that they had produced HPA (section 4.3)? The method is clear, but were there standards used to verify the compound purified? Was mass spec performed? What was the purity?

Answer:

In our experiment, we analyzed HPA by NMR and mass spectrometry. We determined the molecular weight from the mass spectrometry results and compared the NMR data to the literature [Wang, X.J.; et, al. Chemical constituents from Rhodiola wallichiana var. cholaensis (I). Chin. Tradit. Herb. Drugs 2015, 46, 3471-3474.] and determined the structure to be 4-hydroxyphenylacetic acid. The purity of the purified compound HPA was 97.3 % by HPLC analysis.

The images and data of mass spectrometry, nuclear magnetic and HPLC analysis are as follows:

Compound HPA: White crystal powder, ESI-MS m/z: 151 [M – H]. 13C NMR (100 MHZ, CD3OD) δC: 175.0 (C-8), 156.0 (C-4), 129.9 (C-2, 6), 125.6 (C-1), 114.8 (C-3, 5), 39.9 (C-7); 1H NMR (400 MHZ, CD3OD) δH: 7.09 (2H, d, J = 8.8 Hz, H-2, 6), 6.72 (2H, d, J = 8.8 Hz, H-3, 5), 3.47 (2H, s, H-7).

13C NMR

1H NMR

ESI-MS

HPLC analysis

4-hydroxyphenylacetic acid standard

Compound HPA

Remark 10:

Comments to the Author

Section 4.4. So, were three replicates of conditions with 10 larvae per condition the model used? Also, what criteria were used to decide if a fish was healthy or not? What criteria were used to define death?

Answer

In Section 4.4, we adopted a model where each condition was repeated three times, with 10 fish per well and a total of 30 fish per condition. Zebrafish that were observed under a microscope and had no deformities, normal development and normal behavior were considered healthy, and zebrafish whose hearts have stopped beating are considered dead.

Remark 11:

Comments to the Author

Please indicate the vendors for the hardware and software used to conduct these experiments.  Please indicate the dyes used and concentration in the immunofluorescence assays.  Define the characteristics of the inflammatory cells documented.  As written, these experiments cannot be repeated.

Answer:

In lines 274-283, 313-314, 354-355 and 376, we have indicated the suppliers of the hardware and software used for the experiments based on your suggestions.

The zebrafish Tg(zlyz:EGFP) we used in the immunofluorescence assays was a transgenic zebrafish with EGFP labeled macrophages. This causes its macrophages to exhibit green fluorescence when exposed to blue light. It does not require the use of a dye.

In figure 5, The inflammatory cells we documented were macrophages that migrated to the lateral line position after the zebrafish cloaca (The white dashed area in the figure is the cloaca, the position indicated by the yellow arrow is the zebrafish lateral line and the red dashed area is the statistical area).

Remark 12:

Comments to the Author

There is no comprehensive statistical section to present the methods, software or power of the various analyses found in the figures. The survival curve was “calculated.” Software and rationale for methods chosen must be presented.

Answer:

In Section 4.8, lines 400-404, we have added a comprehensive statistical analysis section to describe the analysis methods and software used in the charts. The lethal curve was plotted with by GraphPad Prism 9.0 with the HPA concentration and mortality rate as the horizontal and vertical coordinates, respectively, and then the LC1 was calculated by Nonlinear regression (curve fit), this was explained in lines 314-315.

Thank you very much again for your patience and consideration.

Yours sincerely,

Xiaobin Li

Reviewer 3 Report

Comments and Suggestions for Authors

The manuscript entitled “Marine fungus-derived natural compound 4-hydroxyphenylacetic acid induces autophagy to exert antithrombotic effects in zebrafish” describes the effect of isolated 4-hydroxyphenylacetic acid on zebra fish and the hopes associated with it as an anticoagulant drug.

The main comments:

Generally, the article is written clearly, especial, the introduction and results, but the discussion needs to be improved because for now, it sounds like conclusions.

With minor corrections:

-          an explanation of the abbreviations used should be provided at the point where these abbreviations first appear: page 1 line 28, page 2 line 80, page 8 line 166, page 9 line 174, page 14 line 341;

-          page 3 line 101, Figures 2 B and C are blurry.

Author Response

Dear reviewer,

Thank you very much for all your kind review on our manuscript and for your valuable suggestions.

The comments from you have been concerned in details. According to those helpful comments, we made a careful revision on the original manuscript “Marine fungus-derived natural compound 4-hydroxyphenylacetic acid induces autophagy to exert antithrombotic effects in zebrafish” (Manuscript ID: marinedrugs-2912157). Here we submit you the revised version of the manuscript.

We have managed to minimize all kinds of errors and obscurities in order to make it understood without difficulty. We hope it is not about to disappoint you and will feel very grateful if it is able to fulfill the standards of your journal and be accepted. Anyway, a lot of appreciation will be paid to you for your helpful work, beneficial suggestions and extraordinary patience during this process.

All the changes were listed as follows:

Remark 1:

Comments to the Author

Generally, the article is written clearly, especial, the introduction and results, but the discussion needs to be improved because for now, it sounds like conclusions.

Answer:

In lines 238-252, we have provided a more detailed explanation and discussion of the experimental results, The discussion section has been further improved.

Remark 2:

Comments to the Author

An explanation of the abbreviations used should be provided at the point where these abbreviations first appear: page 1 line 28, page 2 line 80, page 8 line 166, page 9 line 174, page 14 line 341.

Answer:

Based on your suggestion, in lines 28, 181, 189, 377, we have provided corresponding explanations for abbreviations that appear for the first time.In line 85, we used the full name

Remark 3:

Comments to the Author

 page 3 line 101, Figures 2 B and C are blurry

Answer:

In line 105,we have adopted a clearer version of Figure 2.

Thank you very much again for your patience and consideration.

Yours sincerely,

Xiaobin Li

Round 2

Reviewer 2 Report

Comments and Suggestions for Authors

This is a huge improvement compared to the original work. The only outstanding issue is again statistical. What post hoc test (e.g., SNK, Tukey) was used for the multiple comparisons between groups after one-way ANOVA?

Author Response

Dear reviewer,

Thank you very much for all your kind review on our manuscript and for your valuable suggestions.

The comments from you have been concerned in details. According to those helpful comments, we made a careful revision on the original manuscript “Marine fungus-derived natural compound 4-hydroxyphenylacetic acid induces autophagy to exert antithrombotic effects in zebrafish” (Manuscript ID: marinedrugs-2912157). Here we submit you the revised version of the manuscript.

We have managed to minimize all kinds of errors and obscurities in order to make it understood without difficulty. We hope it is not about to disappoint you and will feel very grateful if it is able to fulfill the standards of your journal and be accepted. Anyway, a lot of appreciation will be paid to you for your helpful work, beneficial suggestions and extraordinary patience during this process.

The changes were listed as follows:

Remark 1:

Comments to the Author

This is a huge improvement compared to the original work. The only outstanding issue is again statistical. What post hoc test (e.g., SNK, Tukey) was used for the multiple comparisons between groups after one-way ANOVA?

Answer:

In line 402, we explained that the comparison between groups was performed by One-Way ANOVA and Dunnett's t-test. At p values less than 0.05, the differences between the groups were considered significant.

Thank you very much again for your patience and consideration.

Yours sincerely,

Xiaobin Li